# Impact of Proton Pump Inhibitors and Histamine-2-Receptor Antagonists on Non-Small Cell Lung Cancer Immunotherapy: A Systematic Review and Meta-Analysis

**DOI:** 10.3390/cancers14061404

**Published:** 2022-03-09

**Authors:** Alessandro Rizzo, Antonio Cusmai, Francesco Giovannelli, Silvana Acquafredda, Lucia Rinaldi, Andrea Misino, Elisabetta Sara Montagna, Valentina Ungaro, Mariagrazia Lorusso, Gennaro Palmiotti

**Affiliations:** 1Struttura Semplice Dipartimentale di Oncologia Medica per la Presa in Carico Globale del Paziente Oncologico “Don Tonino Bello”, I.R.C.C.S. Istituto Tumori “Giovanni Paolo II”, Viale Orazio Flacco 65, 70124 Bari, Italy; a.cusmai@oncologico.bari.it (A.C.); f.giovannelli@oncologico.bari.it (F.G.); s.acquafredda@oncologico.bari.it (S.A.); l.rinaldi@oncologico.bari.it (L.R.); dott.misino@oncologico.bari.it (A.M.); es.montagna@oncologico.bari.it (E.S.M.); g.palmiotti@oncologico.bari.it (G.P.); 2S.C. Farmacia e U.Ma.C.A., Istituto di Ricerca e Cura a Carattere Scientifico (IRCCS), Istituto Tumori Giovanni Paolo II-Bari, 70124 Bari, Italy; v.ungaro@oncologico.bari.it; 3Unità Operativa Complessa Chirurgia Toracica, Istituto di Ricerca e Cura a Carattere Scientifico (IRCCS), Istituto Tumori Giovanni Paolo II-Bari, 70124 Bari, Italy; mg.lorusso@oncologico.bari.it

**Keywords:** antiacid, proton pump inhibitors, immunotherapy, non-small cell lung cancer, immune checkpoint inhibitors

## Abstract

**Simple Summary:**

The current meta-analysis highlighted that proton pump inhibitors (PPIs) and histamine-2-receptor antagonists (H2RAs) could impact immune checkpoint inhibitors (ICIs) efficacy in NSCLC patients, highlighting the need for a deeper comprehension of factors involved in treatment response or resistance. Since the number of indications and NSCLC patients receiving ICIs is supposed to increase further soon, identifying the impact of these agents on NSCLC immunotherapy represents a compelling and urgent need regarding NSCLC.

**Abstract:**

(1) Background: In recent years, immunotherapy has revolutionized the treatment landscape of non-small cell lung cancer (NSCLC), representing a therapeutic breakthrough in this field. Antacid agents such as proton pump inhibitors (PPIs) and histamine-2-receptor antagonists (H2RAs) are commonly prescribed for extended periods in NSCLC patients, and these drugs have the potential to modify the efficacy of immune checkpoint inhibitors (ICIs). (2) Materials and Methods: Herein, we conducted a systematic review and meta-analysis to investigate the impact of PPIs and H2RAs on progression-free survival (PFS) and overall survival (OS) among patients receiving immunotherapy for metastatic NSCLC. Effect measures for OS were Hazard Ratios (HRs) and 95% Confidence Intervals (CIs), which were extracted from available studies. Forest plots were used to assess HRs to describe the relationship between treatment and OS in the specified cohorts of patients. (3) Results: Six studies were included in the analysis, involving 2267 patients. The pooled HRs for OS and PFS were 1.4 (95% CI, 1.25–1.58) and 1.29 (95% CI, 1.17–1.43), respectively, suggesting that PPIs and H2RAs administration was negatively associated with PFS and OS. (4) Conclusion: Concomitant antacid use could modify the activity of ICIs in NSCLC patients.

## 1. Introduction

Non-small cell lung cancer (NSCLC) remains one of the most common malignancies worldwide [1]. Several risk factors have been traditionally associated with the onset of NSCLC, including, among others, smoking, air pollution, occupational exposure, radiation, radon, and asbestos [2]. Recent years have witnessed the emergence of a wide range of systemic treatments for NSCLC patients, such as immunotherapy (as monotherapy or in combination with other anticancer agents), and targeted therapies [3]. As regards the former, immune checkpoint inhibitors (ICIs) have revolutionized NSCLC treatment, following the results of several practice-changing phase III clinical trials, and great progress has recently been made in this setting [4]. Firstly, the landmark KEYNOTE-024 phase III study conducted by Reck and colleagues reported the superiority of the PD-1 inhibitor pembrolizumab over standard chemotherapy for NSCLC patients with PD-L1 Tumor Proportion Score (TPS) ≥ 50% [5]. According to the results of this study, ICI monotherapy significantly improved progression-free survival (PFS), overall survival (OS), and overall response rate (ORR) in this patient population, leading to the United States Food and Drug Administration (FDA) approval of this agent for metastatic NSCLC patients without driver gene mutations and PD-L1 expression ≥ 50% [6]. Subsequently, a therapeutic revolution has characterized the NSCLC treatment scenario, as witnessed by the presentation and publication of an impressive number of clinical trials assessing ICIs monotherapy as first- or later-line treatment, as well as immune-based combinations [7]. Among these, the phase III KEYNOTE-189 trial showed a statistically significant and clinically meaningful PFS and OS benefit in non-squamous NSCLC patients treated with first-line pembrolizumab combined with pemetrexed and platinum compared to chemotherapy alone [8]. Similarly, the KEYNOTE-407 reported that a pembrolizumab combination with carboplatin and taxane chemotherapy was superior to chemotherapy alone as front-line therapy in metastatic squamous cell carcinoma [9]. Other immune-based combinations (e.g., chemotherapy plus ICIs and bevacizumab, the double checkpoint blockade with an anti-PD-1 agent and a CLTA-4 inhibitor, etc.) have reported practice-changing results in other trials, including the IMpower130, the CheckMate 227, and the CheckMate 9LA [10,11,12,13,14].

However, if the NSCLC immunotherapy era seems to have come, some questions remain unanswered, including identifying reliable predictors of response and the optimal choice between monotherapy and combination strategies [15,16,17]. Only a part of NSCLC patients seems to benefit from ICIs, and it is fundamental to investigate the underlying mechanisms and factors impairing the efficacy of immunotherapy [18]. For example, recent studies have explored the role of the gut microbiome in affecting physiological immune function, modifying the activity of cancer immunotherapy [19].

Antacid agents such as proton pump inhibitors (PPIs) and histamine-2-receptor antagonists (H2RAs) are commonly prescribed for extended periods in NSCLC patients, and these drugs have been suggested to modify the activity of anticancer therapies through several mechanisms, including gut microbiome changes [20]. However, available literature reports controversial results, with some studies highlighting a trend towards lower ICI activity and worse clinical outcomes in patients receiving antacids and immunotherapy, and other trials reporting no effect or even longer survival in subjects treated with concomitant PPIs or H2RAs [21,22]. Based on these premises, we performed a systematic review and meta-analysis to investigate the impact of antacids on NSCLC patients treated with ICIs.

## 2. Materials and Methods

### 2.1. Search Strategies

All clinical trials published from 10 June 2000 to 15 January 2022, were retrieved. Keywords used for searching on PubMed/Medline, Cochrane Library, and EMBASE were: “immunotherapy” or “nivolumab” or “ipilimumab” or “atezolizumab” or “pembrolizumab” or “durvalumab” or “avelumab” or “immune checkpoint inhibitors” and “metastatic lung cancer” and “lung cancer” or “non-small cell lung cancer” or “NSCLC” AND “proton pump inhibitors” or “PPI” or “omeprazole” or “pantoprazole” or “lansoprazole” or “esomeprazole” or “rabeprazole” or “histamine-2-receptor antagonists” or “ranitidine”. Only articles published in peer-reviewed journals, with available full text, and written in English were considered. Furthermore, proceedings of the main international oncological meetings (American Society of Clinical Oncology, American Association for Cancer Research, European Society of Medical Oncology, European Council of Clinical Oncology) were also searched from 2000 onward for relevant abstracts.

The search and review of the articles were evaluated by 3 authors independently.

### 2.2. Aims of the Systematic Review and Meta-Analysis

The aims of the systematic review and meta-analysis were:To evaluate PFS in NSCLC patients receiving concomitant antacids (PPIs and/or H2RAs) and ICIs.To evaluate OS in NSCLC patients receiving concomitant antacids (PPIs and/or H2RAs) and ICIs.

### 2.3. Selection Criteria

Studies selected from the first analysis were then restricted to: (1) clinical trials in NSCLC patients; (2) participants treated with ICIs; (3) studies with available data in terms of PPIs and H2RAs use; (4) studies with available data regarding PFS and OS in patients receiving immunotherapy.

### 2.4. Data Extraction and Quality Assessment

The following data were extracted for each publication: (1) study information (author, carry out country, inclusion criteria, study design); (2) type and dose of ICI; (3) number of patients; (4) type of antacid (PPIs and H2RAs). Three separate authors conducted the search and identification independently. The current analysis was conducted according to Preferred Reporting Items for Systematic Review and Meta-Analyses (PRISMA) guidelines (Table A1) [23,24]. We state that the current meta-analysis has not been registered on PROSPERO.

### 2.5. Statistical Design

All statistical analyses were performed using ProMeta 3 software.

Effect measures for OS were Hazard Ratios (HRs) and 95% Confidence Intervals (CIs), which were extracted from available studies. Forest plots were used to assess HRs to describe the relationship between treatment and OS in the specified cohorts of patients.

Statistical heterogeneity between trials was examined using the Chi-square test and the I^2^ statistic; substantial heterogeneity was considered to exist when the I^2^ value was greater than 50% or there was a low *p* value (<0.10) in the Chi-square test [25]. When no heterogeneity was noted, the fixed effects model was used, while the random-effects model was applied in the presence of significant heterogeneity.

## 3. Results

### 3.1. Search Results

In our search, we found 1327 potentially relevant reports, which were subsequently restricted to 6 [26,27,28,29,30,31]. We excluded 1321 records as non-pertinent reports (pre-clinical studies, meta-analysis and systematic reviews, review articles, editorials, case reports, ongoing trials/trials in progress, no immunotherapy arm trials), as shown in Figure 1.

We also excluded the study recently published by Hopkins and colleagues since this study included immune-based combinations with chemotherapy and atezolizumab, while our analysis was focused on immunotherapeutic agents used as monotherapy. Table 1. reports a summary of the included studies [26,27,28,29,30,31].

### 3.2. Overall Survival

Six trials included data regarding OS [26,27,28,29,30,31]. The pooled HR for OS was 1.4 (95% CI, 1.25–1.58) (Figure 2), suggesting that patients receiving ICIs and PPIs and/or H2RAs presented lower OS compared to patients without antacids administration; the analysis was associated with low heterogeneity (I^2^ of 0%), and thus, a fixed-effects model was used.

### 3.3. Progression-Free Survival

Five trials included data regarding PFS [27,28,29,30,31]. The pooled HR for PFS in the comparison between UC patients receiving immunotherapy with or without concomitant PPIs and/or H2RAs was 1.29 (95% CI, 1.17–1.43) (Figure 3). The analysis showed low heterogeneity, and a fixed-effect model was used (I^2^ = 0%).

### 3.4. Publication Bias

The funnel plots of OS and PFS showed basic symmetry, suggesting no publication bias (Figure 4 and Figure 5).

## 4. Discussion

Several phase I to III clinical trials have established the role of immunotherapy in metastatic NSCLC, producing unprecedented paradigm shifts in a relatively short period [32,33]. ICIs, as monotherapy or in combination with other anticancer agents, have revolutionized previous NSCLC treatment algorithms, prompting researchers and clinicians to consider the expansion of the role of immunotherapy in other settings, including the earlier stage of the disease (e.g., as neoadjuvant and adjuvant therapy) [34]. To the best of the authors’ knowledge, the current study represents the most comprehensive and updated meta-analysis in literature systematically assessing the impact of concomitant PPIs and/or H2RAs on ICI efficacy in NSCLC. According to our results, the analysis highlighted shorter median OS and median PFS in the case of concomitant antacid administration.

Recent reports have observed the significant importance of gut microbiome in modifying immunotherapy efficacy [35]. Identifying reliable predictors of response to ICIs is needed to better modulate the therapeutic process. Not only PD-L1, TMB, MSI, but also novel, emerging focuses of research are under development, including human microbiota [36,37,38]. In particular, the gut microbiome could be the key to enhancing anticancer immune response and, finally, to improve the prognosis of cancer patients receiving ICIs [39]. In addition, several multicenter, retrospective trials have explored the impact of concomitant medications, including metformin, aspirin, antibiotics, and antacids, on immunotherapy efficacy, reporting conflicting results [40]. Interestingly, the immunomodulatory effect produced by agents such as PPIs could impair the activity of ICIs, modifying gut microbiota—a well-known regulator of homeostasis [41]. From a biological point of view, antacids like PPIs and H2RAs may alter the gut microbiome through several mechanisms, including the decrease of bacterial richness, changes in gastric pH, transformations of bacterial species, and intestinal barrier dysfunctions [42,43]. In addition, some pre-clinical reports have also suggested that PPIs and H2RAs could impair the physiological function of polymorphonuclear neutrophils, natural killer cells, and cytotoxic T-lymphocytes, with all these elements being involved in ICIs efficacy [44,45]. Moreover, growing evidence suggests that gut microbiota dysbiosis may reduce the activity of ICIs, as also suggested in recently published studies in other settings and malignancies, including urothelial carcinoma [46]. For example, a pooled analysis from individual-participant data from IMvigor210 and IMvigor211 has highlighted that PPI use may represent a negative prognostic marker in advanced urothelial carcinoma treated with ICI therapy.

The current meta-analysis presents some strengths and limitations to be noticed. Among the strengths, the study includes an overall large number of metastatic NSCLC patients (*n* = 2267) receiving immunotherapy and represents the most updated meta-analysis on this topic. At the same time, some limitations should be acknowledged. Among these, the meta-analysis was based on aggregate data and not on individual-patient data; secondly, the included studies assessed different ICIs and antacids (PPIs and H2RAs), with these trials reporting also notable differences in terms of study design, sample size, and patient population (e.g., studies conducted in different geographical areas, an element representing a possible source of heterogeneity). Since ICIs such as atezolizumab, nivolumab, and pembrolizumab share some features but do not have superimposable mechanisms of action, this element could have introduced some bias. Lastly, no data regarding the impact of antacids on ICIs toxicity were available, and thus, we did not include this assessment in our meta-analysis. In addition, since it is likely that NSCLC patients included in our analysis were taking not only PPIs or H2RAs but also other medications—and since it is impossible to fully account for these effects—this bias cannot be excluded and avoided. Lastly, since our analysis was focused on ICIs monotherapy, we did not include the recently published study conducted by Hopkins and colleagues since this trial assessed immune-based combinations and not single-agent treatments [32]. Moreover, we did not include patients receiving chemotherapy in our analysis, and there is no control arm with chemotherapy-based regimens.

The current study suggests that concomitant antacids administration could be associated with shorter clinical outcomes in NSCLC patients treated with immunotherapy, confirming the results of recently published retrospective studies. However, most studies are small and underpowered; larger, prospective clinical trials are greatly needed to address this unmet need and identify whether specific medications such as PPIs or H2RAs could affect ICIs efficacy. Although our findings should be interpreted with caution, we believe that the current study has the merit of supporting the exploration of the role of PPIs and H2RAs in NSCLC immunotherapy due to the potentially meaningful clinical impact of these drugs in clinical practice. Further studies are warranted to clarify the relationship between ICIs, PPIs, H2RAs, and gut microbiota.

## 5. Conclusions

The current meta-analysis highlighted that PPIs and H2RAs could impact ICIs efficacy in NSCLC patients, highlighting the need for a deeper comprehension of factors involved in treatment response or resistance. Our results should be interpreted cautiously, and the available evidence is not sufficient to demonstrate a close link between worse clinical outcomes and PPIs and H2RAs use in NSCLC patients treated with ICIs. Since the number of indications and NSCLC patients receiving ICIs is supposed to increase further soon, identifying the impact of these agents on NSCLC immunotherapy represents a compelling and urgent need regarding this aggressive malignancy.

## Figures and Tables

**Figure 1 cancers-14-01404-f001:**
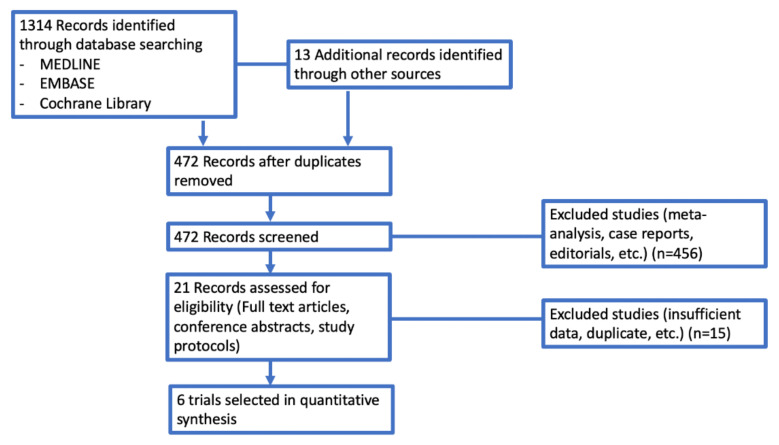
Selection of trials included in the meta-analysis according to PRISMA statement.

**Figure 2 cancers-14-01404-f002:**
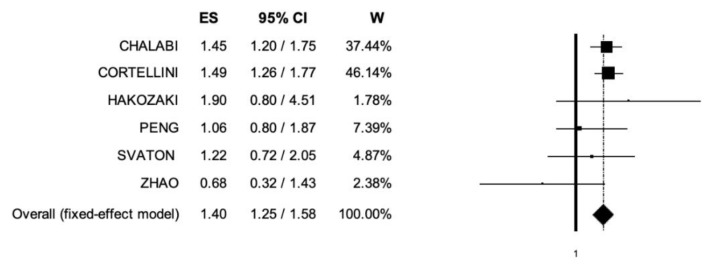
Forest plot of comparison between non-small cell lung cancer patients receiving immune checkpoint inhibitors with concomitant PPIs and/or H2RAs use or not; the outcome (ES) was Hazard Ratio of Overall Survival. Abbreviations: CI: confidence interval; ES: Effect Size (Hazard Ratio); W: Weight.

**Figure 3 cancers-14-01404-f003:**
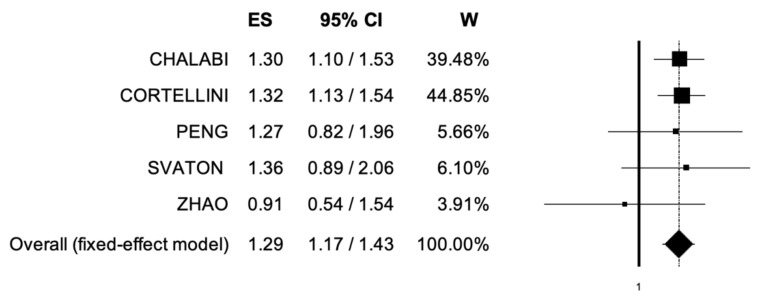
Forest plot of comparison between non-small cell lung cancer patients receiving immune checkpoint inhibitors with concomitant PPIs and/or H2RAs use or not; the outcome (ES) was Hazard Ratio of Progression-Free Survival. Abbreviations: CI: confidence interval; ES: Effect Size (Hazard Ratio); W: Weight.

**Figure 4 cancers-14-01404-f004:**
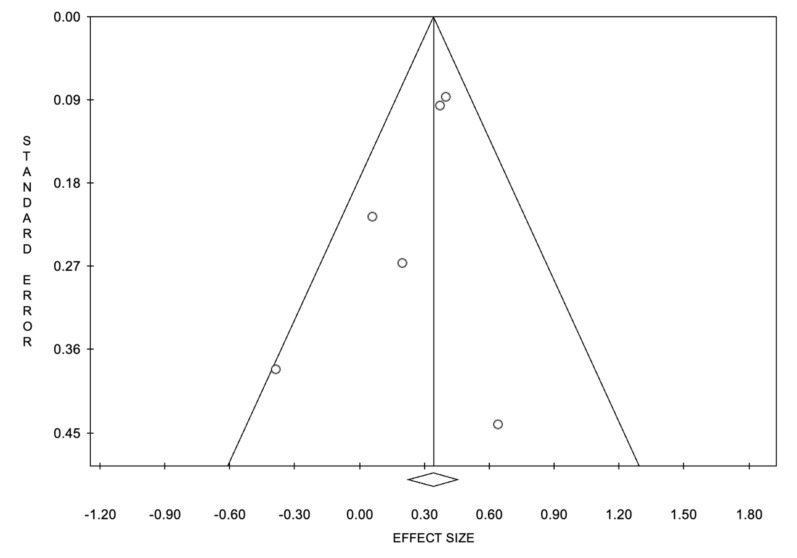
Funnel plot of overall survival.

**Figure 5 cancers-14-01404-f005:**
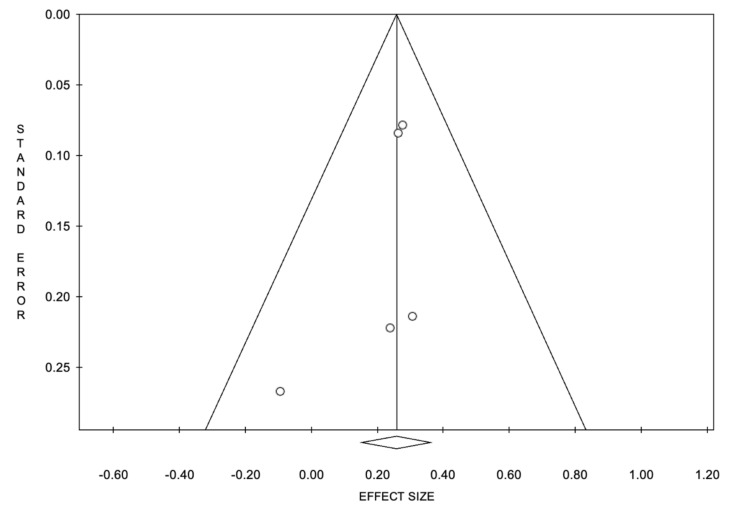
Funnel plot of progression-free survival.

**Table 1 cancers-14-01404-t001:** Summary of all the included studies in the present meta-analysis.

Author Name [Reference]	Year	Region	Number of Patients Receiving Antacids	Number of Patients No Antacids	Type of ICIs	Type of Antiacids
Hakozaki [26]	2019	Japan	47	43	Nivolumab	PPIs or H2RAs
Zhao [27]	2019	China	40	69	Nivolumab Pembrolizumab SHR-1210	PPIs
Chalabi [28]	2020	Worldwide	234	523	Atezolizumab	PPIs
Peng [29]	2021	United States	89	144	Nivolumab Pembrolizumab Nivolumab plus ipilimumab	PPIs
Rounis [30]	2021	Greece	23	43	Atezolizumab Nivolumab Pembrolizumab	PPIs
Cortellini [31]	2021	Italy	547	465	Atezolizumab Nivolumab Pembrolizumab	PPIs or H2RAs

Abbreviations: H2RAs: histamine-2-receptor antagonists; ICIs: immune checkpoint inhibitors; PPIs: proton pump inhibitors

## Data Availability

Not applicable.

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
