# Peer review of "Impact of Proton Pump Inhibitors and Histamine-2-Receptor Antagonists on Non-Small Cell Lung Cancer Immunotherapy: A Systematic Review and Meta-Analysis"

_cancers, 2022, doi:10.3390/cancers14061404_

Round 1

Reviewer 1 Report

The manuscript is short and aims to highlight a topical point of interest – Do proton pump inhibitors (PPIs) and histamine-2-receptor antagonists (H2RAs) modify the efficacy of immune checkpoint inhibitors (ICIs) in NSCLC? Manuscript aims to conclude - Concomitant antacid use could impact ICIs efficacy in NSCLC patients. This is based on a finding that PPIs and H2Ras were prognostically associated with worse OS and PFS.

A major limitation is that there is no analysis of analyses of patients treated with chemotherapy. This limitation should be clearer to this. Conclusions should not be using the word efficacy.

The discussion should be updated to include data from other cancer, particularly if the data has two arms: e.g. https://doi.org/10.1158/1078-0432.CCR-20-1876 (Concomitant Proton Pump Inhibitor Use and Survival in Urothelial Carcinoma Treated with Atezolizumab)

Most importantly studies were not retrieved out to January, 15 2022. I would suggest updating the manuscript. Below is an example of a manuscript including >800 patient with NSCLC treated with atezolizumab (and 400 treated with chemotherapy) in which the impacts of PPIs have been examined: https://doi.org/10.1038/s41416-021-01606-4 (Efficacy of first-line atezolizumab combination therapy in patients with non-small cell lung cancer receiving proton pump inhibitors: post hoc analysis of IMpower150)

There may be more articles.

Author Response

Dear Reviewer, 

Thank you for the time spent revising our work.

As you could find in the Revised Manuscript, we updated the analyses and we modified several sections of the paper, as required.

We hope the revised manuscript will better suit the journal.

Reviewer 2 Report

Comments for the authors:

In the manuscript entitled “Impact of Proton Pump Inhibitors and Histamine-2-Receptor Antagonists on Non-Small Cell Lung Cancer Immunotherapy: A Systematic Review and Meta-Analysis", Alessandro Rizzo et al, comprehensively reviewed and conducted meta-analysis to investigate the impact of PPIs and H2RAs on PFS and OS among patients receiving immunotherapy for metastatic NSCLC. This study suggests that clinicians should take into consideration the negative influence of antacid agent on the efficacies of ICI. Although this research is very interesting and addresses important topic, there are some concerns to be solved. The authors are required to answer the following questions and improvements should be implemented in this manuscript.

Comments:

  • Figure 2 and Figure 3: Please confirm “ES; effect size” in figures is appropriate term. It seems “Hazard ratio” (HR) is more appropriate.

  • The legend of Figure 3: The authors describe “overall survival”, but it seems “PFS” is appropriate. Please check Figure 2 and Figure 3, and also their figure legends.

Author Response

Dear Reviewer, 

Thank you for the time spent revising our manuscript.

As you could find in the Revised Manuscript, we modified legends of figures (orange) and we updated the analysis also adding a recent study. 

We hope the revised paper will better suit the journal. Thank you again.

Round 2

Reviewer 1 Report

The updates are appreciated. As previous a major limitation is that there is no analysis of analyses of patients treated with chemotherapy. This limitation should be clearer to this.

The below sentence should be updated to indicate that as there is no control arm no insight to treatment effect are provided, only the prognostic:

Moreover, we did not include patients receiving chemotherapy in our analysis.

Author Response

Dear Reviewer,

Thank you again for the time spent revising our work.

We included the suggested sentence in the Revised Manuscript.

Thank you again.